# Study of the Structural and Robustness Characteristics of Madrid Metro Network

**Elisa Frutos Bernal** [1,*,†] and **Angel Martín del Rey** [2,†]

1 Department of Statistics, University of Salamanca, 37008 Salamanca, Spain
2 Department of Applied Mathematics, Institute of Fundamental Physics and Mathematics, University of Salamanca, 37008 Salamanca, Spain; delrey@usal.es
* Correspondence: efb@usal.es
† These authors contributed equally to this work.

**Abstract:** A transportation service must be sustainable, respectful of the environment, and socially and economically responsible. These requirements make metro networks the ideal candidate as the most efficient mean of transport in our society. Now, a correct management of this type of infrastructures entails the analysis of the structure and robustness of these networks. This allows us to detect malfunctions and, above all, to design in the most appropriate way the expansion of subway networks. This is one of the major challenges facing the study of transport networks in sustainable smart cities. In this sense, the complex network analysis provides us with the necessary scientific tools to perform both quantitative and qualitative analysis of metro networks. This work deals with Madrid metro network, which is the largest in Spain. The main structural and topological characteristics, and robustness features of Madrid metro network were studied. The results obtained were analyzed and some conclusions were derived.

**Keywords:** complex network analysis; subway networks; centrality measures; robustness; Madrid metro network

## 1. Introduction

One of the central pillars on which sustainability is based is the design and implementation of procedures that allow us to manage in an efficient way the public resources, services and infrastructures. Among them, public transport systems and, more precisely, urban rail transit systems, constitute the paradigmatic example in which sustainability and development must coexist efficiently.

The design of efficient public urban transportation plans has a huge social, economic and environmental impact. Urban transport networks (bus networks, subway networks, light rail networks, etc.) are a backbone of society and one of the pillars on which the paradigm of sustainable smart cities is sustained. Currently, one of the major concerns is the efficient management of such transportation networks. As these infrastructures can be mathematically modeled in terms of graphs, the complex network analysis plays a very important role in their study. Usually, *L*-space configuration is used when representing metro networks as graphs. In this case, stations stand for the nodes of the graph and direct rail connections between them define the edges. Depending on the role played by the associated station, every node of the network can be classified into three classes (which are not necessarily disjoint compartments): monotonic nodes, transfer nodes and termini nodes. Monotonic nodes are those that belong to only one metro line, transfer nodes belong to (at least) two lines of the metro network, and finally termini nodes are the end stations of each line. Transfer and termini nodes are also called diatonic nodes.

There are alternative ways to represent metro networks in terms of graph theory. For example, the reduced *L*-space configuration is obtained from the standard *L*-space when the monotonic

non-transfer stations are removed. The *P*-space representation is obtained when the nodes of the graph stand for the stations and there exists an edge between two nodes when these nodes belong to the same metro line. In addition, in the *C*-space configuration, the nodes represent the subway lines and there exists an edge between two nodes if the associated lines share, at least, one station.

This work tackles the analysis of a subway network. Metro networks have a characteristic that distinguish them from other urban transport networks—such as bus networks: it is practically impossible to restructure the already built network. Consequently, the knowledge about the structural properties and robustness characteristics of such transport networks play a fundamental role in the optimization of resources (both, material and economic) for its efficient maintenance. In this way, it is necessary to identify the stations and links of major structural importance within the whole network and whose malfunction modifies its robustness. Furthermore, this analysis is also extremely interesting when the expansion of metro network is tackled. We focus our attention on Madrid metro network, which is the largest in Spain not only because of the number of stations (243) but also because of its total length (294 km) and the number of passengers (during 2017, 626.4 million people used it—2.3 million on working days—and about 687 million passengers in 2018). Madrid metro network will face in the coming years its remodeling and expansion (due to the urban growth). Consequently, it is mandatory to have studies and analysis for making decisions about the efficient and sustainable management of the existing network and the location of new stations.

The mathematical analysis of transport networks is an old issue [1], although it has gained great popularity and attention in the last years. In these studies, the complex network analysis plays an important role [2] and they focus the attention not only on subway networks but also on bus networks [3,4] and on road networks [5,6]. In the case of subway networks, several works have been published in the scientific literature. Some of them study specific characteristics: for example, in [7], the analysis of the betweenness centrality associated to the reduced *L*-space topology of 32 metro networks in the world is shown; the analysis performed in [8] reveals two related classes of complex networks that can be approximated by an evolutionary complex network with an associated degree distribution; or, in [9], the authors proposed an alternative methodology (different from *L*-space or *P*-space approaches) to represent the topology of subway networks and the main coefficients are computed and compared in the case of Nanjing (China). Furthermore, the analysis of robustness of metro networks against different type attacks has also been presented in some works: for example, the transport capacity and the local and global connectivity of Shanghai, Beijing and Guangzhou metro networks are computed and analyzed in [10]; in [11], some centrality measures considering the passengers flow are analyzed considering relative disruption probability of each subway line and this study is applied in the case of Shanghai metro network; etc.

As far as we know, there is no detailed study on the structural and robustness characteristics of Madrid metro network considering the *L*-space topology. This is precisely the main goal of this work and one of the most important contributions apart from the novel study of the robustness of the metro network considering as a basis the notion of metro lines. In this paper, the most important centrality measures, structural coefficients and robustness indicators of this subway network are analyzed. These provide us sufficient knowledge to draw conclusions about the structural robustness of Madrid metro network.

The rest of the paper is organized as follows. In Section 2, the basics of complex network analysis are shown, the computation and the analysis of the main structural coefficients and parameters of Madrid metro subway are introduced in Section 3. Section 4 is devoted to the study of its robustness. Finally, the conclusions and further work are presented in Section 5.

## 2. The Basics of Complex Network Analysis

In this study, we considered the *L*-space representation of the network, thus the stations of the subway network were represented by nodes of a graph and the tracks connecting two stations were represented by edges of the graph. Therefore, the subway network was represented by a undirected

graph $G = (V, E)$ where $V = \{v_1, v_2, \ldots, v_N\}$ is the set of nodes (note that $v_j$ represents the $j$th node/station of the metro network), and $E = \{e_{ij} = (v_i, v_j), v_i, v_j \in V\}$ is the set of edges, where $|E| = M$.

The adjacency matrix of $G$, $A_G = (a_{ij})_{1 \leq i,j, \leq N}$, is a $N \times N$ symmetric matrix such that the coefficient $a_{ij}$ takes the value 1 or 0 depending on whether there is a link between nodes $v_i$ and $v_j$. The *degree* of a node $v_i$ is the number of adjacent nodes to $v_i$ and can be computed as follows: $k_i = \sum_{j=1}^{N} a_{ij}$.

The Laplacian matrix of $G$, $Q_G = \Delta - A_G$, is an $N \times N$ matrix where $\Delta = \text{diag}(k_1, \ldots, k_N)$ is the $N \times N$ diagonal degree matrix. As shown in Section 2.2, the eigenvalues of $Q_G$ play a very important role in robustness analysis; they are non-negative and can be ordered and denoted as follows: $0 = \lambda_N \leq \lambda_{N-1} \leq \ldots \leq \lambda_1$.

## 2.1. Centrality Measures

The analysis of a complex network is performed through the computation and analysis of several structural coefficients of the network topology. Specifically, the most important are the following [12]: degree centrality, average degree, degree distribution, average path length, closeness centrality, and betweenness centrality. Some of the metrics and coefficients introduced are local—related to nodes—and others are global—related to the whole network.

### 2.1.1. Local Measures and Coefficients

The *degree centrality* of $v_i$ is the average number of incident edges to $v_i$, that is:

$$C_D(v_i) = \frac{k_i}{N-1}, \quad 0 \leq C_D(v_i) \leq 1. \tag{1}$$

The *shortest path length* or *distance* between two nodes $v_i, v_j \in V$ is denoted by $d(v_i, v_j)$ and it is defined as the minimum number of links necessary to follow from node $v_i$ to node $v_j$.

The *eccentricity* of the $i$th node $v_i$ is defined as the maximum distance from $v_i$ to another node of the network: $e(v_i) = \max\{d(v_i, v_j), 1 \leq j \leq N, j \neq i\}$.

The *farness centrality* of the $i$th node is defined as the sum of its distances to all other nodes of the network:

$$C_F(v_i) = \sum_{l=1, l \neq i}^{N} d(v_i, v_l), \tag{2}$$

whereas the *closeness centrality* of $v_i$ is the inverse of its farness: $C_{CL}(v_i) = \frac{1}{C_F(v_i)}$, where $\frac{2}{(N-1)N} \leq C_{CL}(v_i) \leq \frac{1}{N-1}$. Note that the greater is the value of closeness centrality, the smaller is the length of the shortest paths to all other nodes and the more central the node is. The *normalized closeness centrality* is obtained by multiplying by $N-1$, that is: $\frac{2}{N} \leq \widetilde{C}_{CL}(v_i) = (N-1)C_{CL}(v_i) \leq 1$.

Finally, the *betweenness centrality* of the node $v_i \in V$ is defined mathematically as follows:

$$C_B(v_i) = \sum_{r \neq s \neq i} \frac{\ell_{rs}(v_i)}{\ell_{rs}}, \tag{3}$$

where $\ell_{rs}$ is the total number of shortest paths from $v_r$ to $v_s$, and $\ell_{rs}(v_i)$ is the the number of shortest paths between $v_r$ and $v_s$ that pass through $v_i$. Note that this centrality index "measures" in some way the number of shortest paths between two nodes that run through a fixed node. In addition, the greater is the number of paths that pass through a node, the greater is the importance of this node and more central it is. Betweenness centrality can be normalized obtaining the *normalized betweenness centrality* $\widetilde{C}_B$ by dividing by $(N-1)(N-2)/2$.

### 2.1.2. Global Measures and Coefficients

The *average degree* of the network $G$ is defined as the average value of all node degrees:

$$\langle k \rangle = \frac{1}{N} \sum_{i=1}^{N} k_i = \frac{N-1}{N} \sum_{i=1}^{N} C_D\left(v_i\right), \quad 0 \le \langle k \rangle \le N-1. \tag{4}$$

Consequently, the *normalized average degree* of $G$ is $\frac{\langle k \rangle}{N-1}$. On the other hand, the *degree distribution* of the network, $P(k)$, is the probability distribution of degrees over the whole network. Note that $\langle k \rangle = \sum_{k=1}^{N} kP(k)$. Moreover, the *cumulative degree distribution* $P_k$ stands for the probability that the degree of a node chosen at random is, at least, $k$.

The *density* of the network $G$ is the ratio between the number of existing edges and the total number of possible edges in the graph, that is:

$$d = \frac{2M}{N(N-1)} = \frac{\langle k \rangle}{N-1}, \quad 0 \le \mu \le 1, \tag{5}$$

which is equal to the normalized average degree.

Taking advantage of the definition of shortest path length, the notion of *diameter D* of the network $G$ can be introduced as the greatest distance between any pair of nodes: $D = \max\{d\left(v_i, v_j\right), 1 \le i < j \le N\}$. Obviously, $D = \max\{e(v), v \in V\}$. Furthermore, the *average path length* of the network is the average distance between any two nodes:

$$L = \frac{2}{N(N-1)} \sum_{1 \le i < j \le N} d\left(v_i, v_j\right), \quad 1 \le L \le D. \tag{6}$$

The *center* of the network $G$, $c(G)$, is the set of vertices with minimal eccentricity. This minimal eccentricity is called *radius* of the network $r(G)$.

### 2.2. Robustness Metrics

Robustness can be defined as the network's ability to survive random failures or deliberate attacks consisting of the elimination of nodes and/or edges [13]. That is, this characteristic refers to the capacity of the network to solve possible failures by offering alternative routes that overcome the attacked edges or nodes; this is a very important issue when metro networks are analyzed [14]. In this sense, several theoretical and numerical robustness metrics have been proposed to quantitatively determine this characteristic. In what follows, a brief description of the most important robustness measures is introduced.

### 2.2.1. Theoretical Robustness Metrics

One of the most important robustness parameters is the *assortativity coefficient*:

$$r = \frac{\sum\limits_{1 \le i,j \le N} \left(a_{ij} - \frac{k_i k_j}{2M}\right) f_{ij}}{\sum\limits_{1 \le i,j \le N} \left(k_i \delta_{ij} - \frac{k_i k_j}{2M}\right) f_{ij}}, \tag{7}$$

where $a_{ij}$ is the corresponding adjacency matrix entry, and $\delta_{ij} = 1$ if there exists an edge between $v_i$ and $v_j$ and 0 otherwise. This coefficient measures how high-degree (respectively, low-degree) nodes are, on average, linked to other nodes with high-degree (respectively, low-degree) [15]. It is defined as the Pearson correlation between the degree of nodes of each edge in the network and ranges from $-1$ (when low-degree nodes are usually connected to high-degree nodes: the network is disassortative) to 1 (nodes with equal or similar degree are often linked: the network is assortative). Assortative

networks are more robust since the attack against a high-degree node could leave the path length relative intact.

A measure of robustness of metro networks needs to take into account the number of alternative routes between two nodes. The more alternative routes there are, the more robust the network is. The *robustness indicator* $r^T$ accounts for the probability of failures/accidents, which is highly dependent of the size of the network. It uses the *cyclomatic number* $\mu$ [16] to calculate the number of paths available in a graph: $\mu = M - N + P$, where $P$ is the number of subgraphs. Therefore, this coefficient is originally defined as follows [17]:

$$r^T = \frac{\mu - M^m}{N},$$ (8)

where $M^m$ is the number of multiple links between two nodes. Subway networks are usually connected ($P = 1$), thus:

$$r^T = \frac{M - N + 1 - M^m}{N}.$$ (9)

Note that $0 \le r^T \le \frac{N^2 - 3N + 2 - 2M^m}{2N}$ and $r^T$ increases when there are alternative paths to reach a destination, and it decreases in large systems.

The *effective graph resistance*, $R_G$, gives us an idea of the robustness of the network by studying the number of parallel paths between two nodes and their length. In this sense, a small value of $R_G$ means that the network is robust. Mathematically it is determined by:

$$R_G = N \sum_{i=1}^{N-1} \frac{1}{\lambda_i},$$ (10)

where $\lambda_i$ is the $i$th eigenvalue of the Laplacian matrix of the graph. In this work, we use the normalized version of the effective graph resistance, which is called *effective graph conductance* defined as follows [18]:

$$C_G = \frac{N-1}{R_G}.$$ (11)

Note that $0 < C_G \le 1$ and, the higher the effective graph resistance is, the higher the robustness of a network is.

*The average efficiency* $E_G$ is a measure that indicates the capability of the network to permit movement between any pair of nodes. Mathematically, it is defined as the averaged sum of the reciprocal of the distances between nodes [19]:

$$E_G = \frac{2}{N(N-1)} \sum_{1 \le i < j \le N} \frac{1}{d(v_i, v_j)}, \quad 0 < E_G \le 1.$$ (12)

Note that the higher its value is, the greater is the robustness of the network.

The *clustering coefficient* [20] is used to assess how the neighbors of a node $v_i$ are connected with another. It is explicitly defined as follows:

$$C_{CLU}(v_i) = \frac{2\epsilon_i}{k_i(k_i - 1)}, \quad 0 \le C_{CLU}(v_i) \le 1,$$ (13)

where $\epsilon_i$ is the number of links connecting neighbors of node $v_i$. From this definition, it follows that the larger the clustering coefficient is, the better the local connectivity around the node $v_i$ is. A global coefficient, called *average clustering coefficient*, can be obtained from this local measure. It is depicted as follows:

$$\tilde{C}_{CLU} = \frac{1}{N} \sum_{i=1}^{N} C_{CLU}(v_i), \quad 0 \le \tilde{C}_{CLU} \le 1.$$ (14)

These clustering coefficients measure the connectivity of a network since they assess how the neighbors of a node are connected with another one [12].

The *algebraic connectivity* is a measurement of the connectivity level of the network. This coefficient is given by the second smallest eigenvalue of the Laplacian matrix, $\lambda_{N-1}$, and it satisfies the following:

$$0 \leq \lambda_{N-1} \leq \kappa_V \leq \kappa_E \leq \min\{k_1, k_2, \ldots, k_N\}, \tag{15}$$

where $\kappa_V$ and $\kappa_E$ are the *vertex connectivity* and the *edge connectivity*, respectively (minimal number of vertices and edges to be removed to disconnect the network). It can be shown that the higher the algebraic connectivity is, the more difficult it is to disintegrate the network into components and, therefore, the network is more robust [18]. The *normalized algebraic connectivity* is obtained dividing the algebraic connectivity by the total number of nodes: $\widetilde{\lambda}_{N-1} = \frac{\lambda_{N-1}}{N}$.

The *natural connectivity* characterizes the redundancy of alternative paths by quantifying the weighted number of closed walks of all lengths: $S = \sum_{k=1}^{\infty} \frac{l_k}{k!}$, where $l_k = \sum_{i=1}^{\infty} \lambda_i^k$ is the number of closed walks of length $k$. Thus, $S = \sum_{i=1}^{N} e^{\lambda_i}$ and the natural connectivity is then obtained by scaling $S$ [21]:

$$\lambda = \ln\left[\frac{1}{N} \sum_{i=1}^{N} e^{\lambda_i}\right], \quad 0 \leq \lambda \leq \ln\left(\frac{N-1}{e} + e^{N-1}\right) \approx N - \ln N. \tag{16}$$

It changes strictly monotonically with the addition or deletion of edges, thus it is sensitive even to a single link failure. We normalize the natural connectivity dividing by the maximum natural connectivity $N - \ln N$:

$$\widetilde{\lambda} = \frac{\ln\left[\frac{1}{N} \sum_{i=1}^{N} e^{\lambda_i}\right]}{N - \ln N}, \quad 0 \leq \widetilde{\lambda} \leq 1. \tag{17}$$

The higher the natural connectivity is, the higher the robustness of the network is.

Finally, the *percolation limit* $p_c$ is another connectivity measure that computes the critical fraction of nodes that are necessary to be removed from the network in order to disconnect it. This coefficient is defined as follows:

$$p_c = \max\left\{0, 1 - \frac{1}{\kappa - 1}\right\}, \quad 0 \leq p_c < 1, \tag{18}$$

where the quotient $\kappa = \frac{\langle k^2 \rangle}{\langle k \rangle}$ is called *degree diversity*, and $\langle k^2 \rangle = \frac{1}{N} \sum_{i=1}^{N} k_i^2$. It has been shown that, the higher $p_c$ is, the more nodes have to be removed from the network to disintegrate it [22], which means the network is more robust.

### 2.2.2. Numerical Robustness Metrics

Numerical robustness metrics are obtained through simulations so that nodes are removed from the network one by one until the network collapses. This paper considers three strategies for node removal: (i) random node removal; (ii) degree-based node removal; and (iii) betweenness-based removal:

(i)   *Random removal*: The node to be removed is chosen randomly from among all the nodes in the network with equal probability.

(ii)  *Degree-based removal*: The node with the highest degree is removed from the network first, then the highest degree is recalculated and the removals continue.

(iii) *Betweenness-based removal*: The node with the highest betweenness centrality is first deleted from the network, then highest betweenness centrality is recalculated after the removal and the removals continue.

Several topological parameters can be used to illustrate the performance changes of the network. In our case, we use the largest connected cluster (LCC) [23,24], the network efficiency, the average betweenness and the number of non-contact islands.

The largest connected cluster can be defined as follows

$$LCC = N'/N,　　　　　　　　　　　　　　(19)$$

where $N'$ is the number of nodes of the largest connected component after removals.

In this case, the robustness curve is obtained when we represent the size of the largest connected component for an interval of removed nodes $[1, N]$. From it, the critical thresholds $f_{90\%}$ and $f_c$ can be obtained. The critical threshold $f_{90\%}$ is the fraction of nodes that have to be removed from the network so that the largest connected component of the resulting network contains 90% of the original network. In the same way, the critical threshold $f_c$ is defined as the fraction of nodes that have to be removed so that the largest component has only one node. In the case of random node removal, we performed 1000 simulation runs to get these values.

### 2.3. Applications of Complex Network Analysis

Complex network analysis is a very useful discipline in the analysis of several and different phenomena that can be modeled by means of a network. It provides not only numerical metrics but also methods and techniques to understand the topological structure of the network and their possible vulnerabilities [25–27]. Specifically, the main objective of all these procedures is to determine the role played in the complex system by the actors and relations that represent nodes and edges.

Apart from the analysis of transportation networks mentioned in the Introduction, the study of the reliability of power grids is one of the paradigmatic examples of the application of complex network analysis [28]. Several works have appeared in the scientific literature studying the vulnerability of different power grids [29,30] and their topological characteristics [31].

Due to their features, communication networks are also susceptible to be analyzed from this point of view [32]. In this case, several aspects of this type of networks are studied using complex network analysis: pinning analysis [33], mobile phone companies studies [34], etc.

Obviously, several applications to biology, epidemiology and related disciplines can also be found (see, for example, [35–37]).

Furthermore, due to the importance of industrial supply networks in our society, it is also very interesting to study these complex systems using techniques from complex network analysis. In this sense, the most relevant works can be found in [38–40].

### 3. Structural Analysis of Madrid Metro Subway

### 3.1. General Considerations

The Madrid metro is a rapid transit system in Madrid (Spain), which actually consists of 13 operating lines (see Figure 1): Line 1 (Pinar de Chamartín-Valdecarros), Line 2 (Las Rosas-Cuatro Caminos), Line 3 (Villaverde Alto-Moncloa), Line 4 (Argüelles-Pinar de Chamartín), Line 5 (Alameda de Osuna-Casa de Campo), Line 6 (Laguna-Lucero), Line 7 (Hospital de Henares-Pitis), Line 8 (Nuevos Ministerios-Aeropuerto T-4), Line 9 (Paco de Lucía-Arganda del Rey), Line 10 (Hospital Infanta Sofía-Puerta del Sur), Line 11 (Plaza Elíptica-La Fortuna), Line 12 (Puerta del Sur-San Nicasio), and Line R (Ópera-Príncipe Pío). Lines 6 and 12 are circular, and in this study the three lines of the light rail network (ML1: Pinar de Chamartín-Las Tablas; ML2: Colonia Jardín-Estación de Aravaca; and ML3: Colonia Jardín-Puerta de Boadilla) are not considered. The "Cercanías" RENFE commuter rail service (that connects Madrid with its metropolitan area and other towns near Madrid) is also not taken into account.

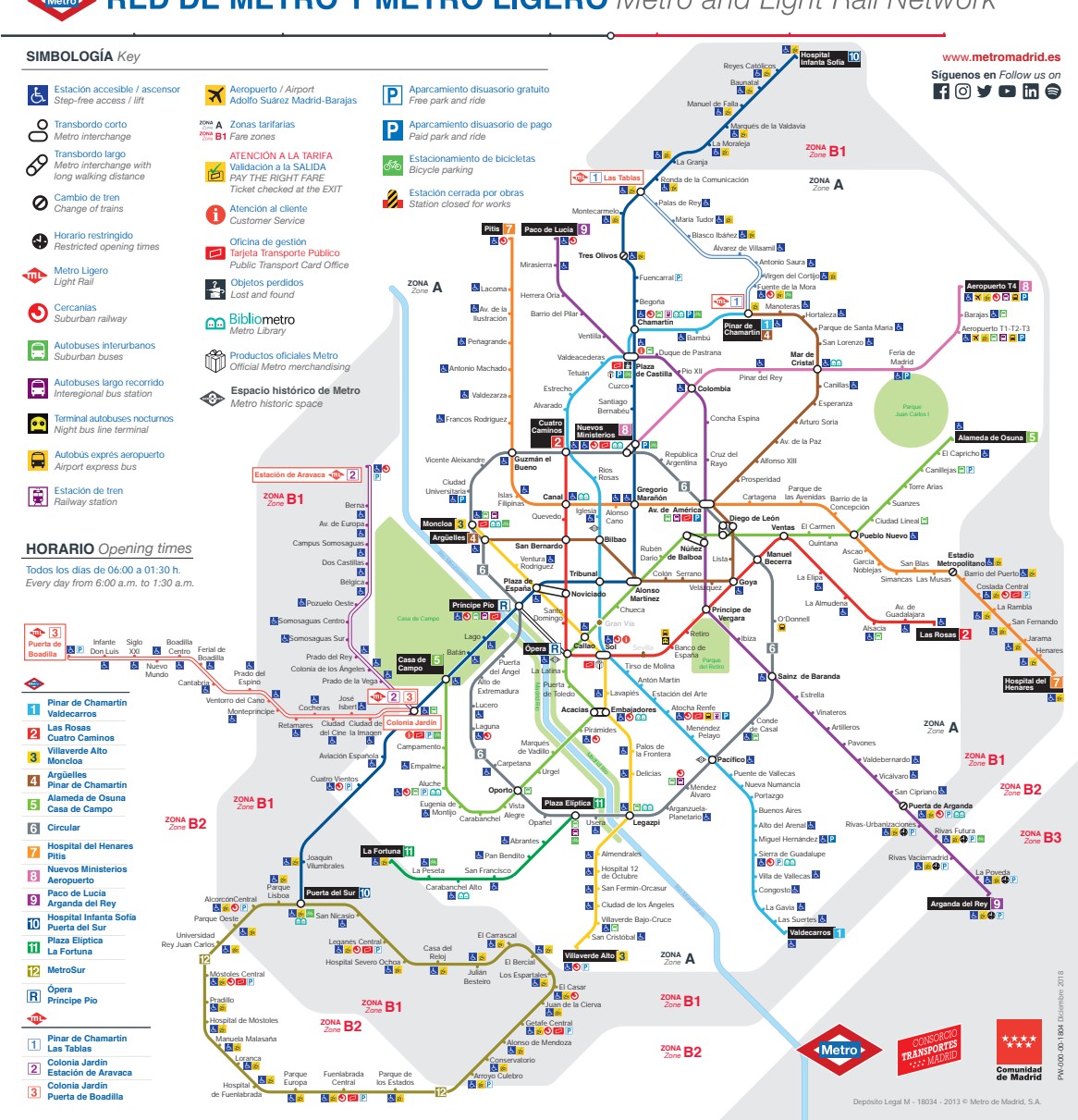

**Figure 1.** The 2019 Madrid metro and light rail networks (courtesy of Metro de Madrid, S.A.).

In this work, the representation of the subway metro follows the *L*-space topology where each station stands for a node of the graph and the edges are defined by means of the direct connections by rail ways between the stations. The number of nodes is $n = 243$ (in Table 1 the types of nodes of each line are shown) and the number of edges is $m = 280$; consequently, the density of the metro network is $d \approx 0.009421$. Note that, for example, the densities of the subway networks of Shanghai, Beijing and Guangzhou—all in China—are 0.0092, 0.0172 and 0.0266 [10]; as a consequence, Madrid metro network is similar to Shanghai. In Figure 2, the graph corresponding to Madrid metro network is shown (the exact spatial location of the stations is not considered). In this figure, cyan nodes are non-transfer termini nodes, and black nodes are transfer nodes. On the other hand, the color code for lines follows those introduced in Table 1.

**Table 1.** Types of nodes of each line of Madrid metro network.

| Line | Monotonic | Transfer (Not Termini) | Transfer (Termini) | Termini (Not Transfer) |
|------|-----------|------------------------|--------------------|------------------------|
| Line 1 (light blue) | 23 | 8 | 1 | 1 |
| Line 2 (red) | 9 | 8 | 1 | 1 |
| Line 3 (yellow) | 10 | 6 | 1 | 1 |
| Line 4 (brown) | 14 | 7 | 2 | 0 |
| Line 5 (green) | 20 | 7 | 1 | 1 |
| Line 6 (gray) | 14 | 14 | - | - |
| Line 7 (orange) | 22 | 5 | 0 | 2 |
| Line 8 (pink) | 4 | 2 | 1 | 1 |
| Line 9 (purple) | 21 | 6 | 1 | 1 |
| Line 10 (blue) | 20 | 9 | 1 | 1 |
| Line 11 (green) | 5 | 0 | 1 | 1 |
| Line 12 (light green) | 27 | 1 | - | - |
| Line R | 0 | 0 | 2 | 0 |

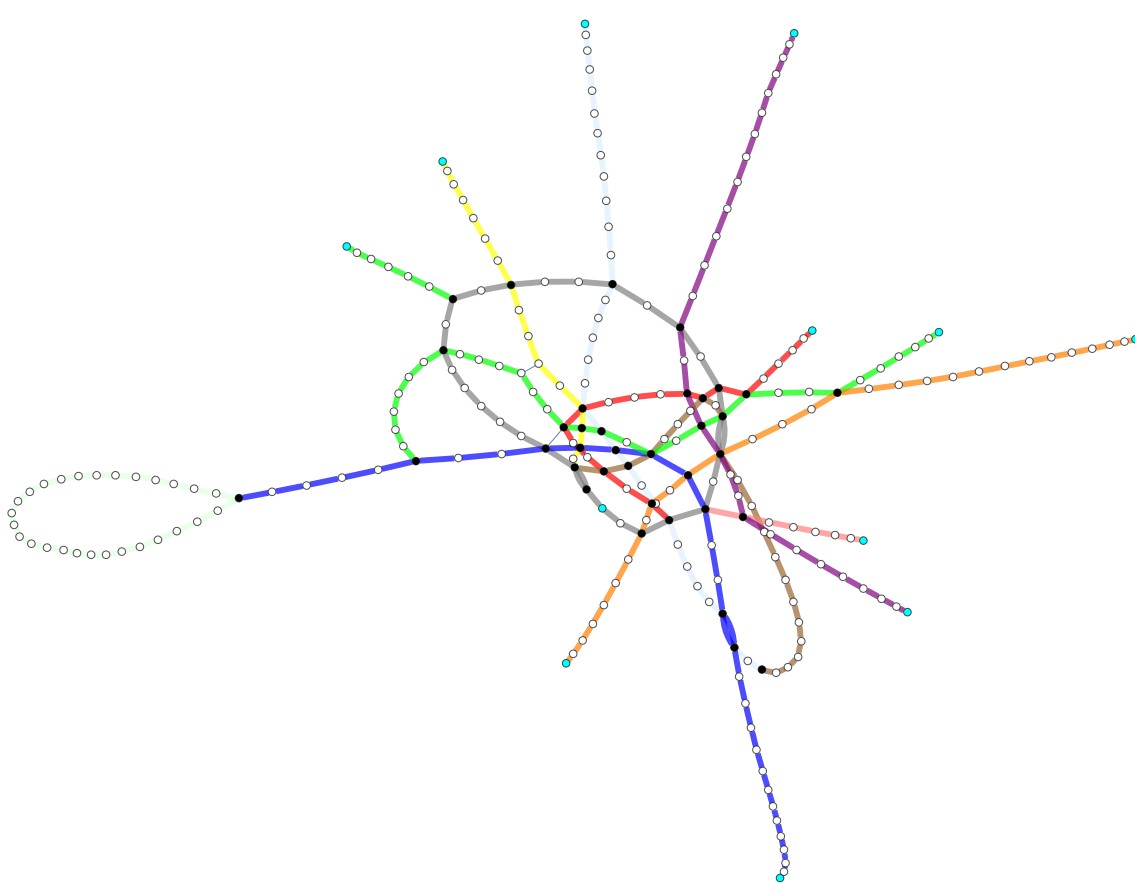

**Figure 2.** The graph representing the Madrid metro network computed using Mathematica.

## 3.2. Basic Study

We computed the most usual coefficients of Madrid metro network used in the complex network analysis to determine the structural importance of each node within the network.

In Table 2, the ten stations with highest degree (and centrality degree) are displayed. In Figure 3, the first five most central stations are shown in the network. As is shown, the most connected node is "Av. de América" with degree 7. Although this station belongs to four lines, it has a double connection with "Diego de León" station and only one of these is taken into account in the computation of its degree. The average degree of the network is $\langle k \rangle \approx 2.280$ and the degree distribution $P(k)$ is shown in Figure 4a, whereas the cumulative degree distribution is introduced in Figure 4b. The fitting function of

the cumulative degree distribution is $h(x) = h_1 e^{h_2 x}$, where $h_1 = 2.012$ and $h_2 = -0.5940$. It is similar to those obtaining from other metro networks around the world such as Shanghai, New York or St. Petersburg [24].

**Table 2.** The ten stations with the highest degree and centrality degree.

| Station | Subway Lines | Degree | Degree Centrality |
|---|---|---|---|
| Av. de América | 4, 6, 7, 9 | 7 | 0.02892 |
| Alonso Martínez | 4, 5, 10 | 6 | 0.02248 |
| Sol | 1, 2, 3 | 6 | 0.02248 |
| Nuevos Ministerios | 6, 8, 10 | 5 | 0.02066 |
| Príncipe Pío | 6, 10, R | 5 | 0.02066 |
| Diego de León | 4, 5, 6 | 5 | 0.02066 |
| Plaza de España | 3, 10 | 5 | 0.02066 |
| Opera | 2, 5, R | 5 | 0.02066 |
| Cuatro Caminos | 1, 2, 6 | 5 | 0.02066 |
| Colombia | 8, 9 | 4 | 0.01653 |

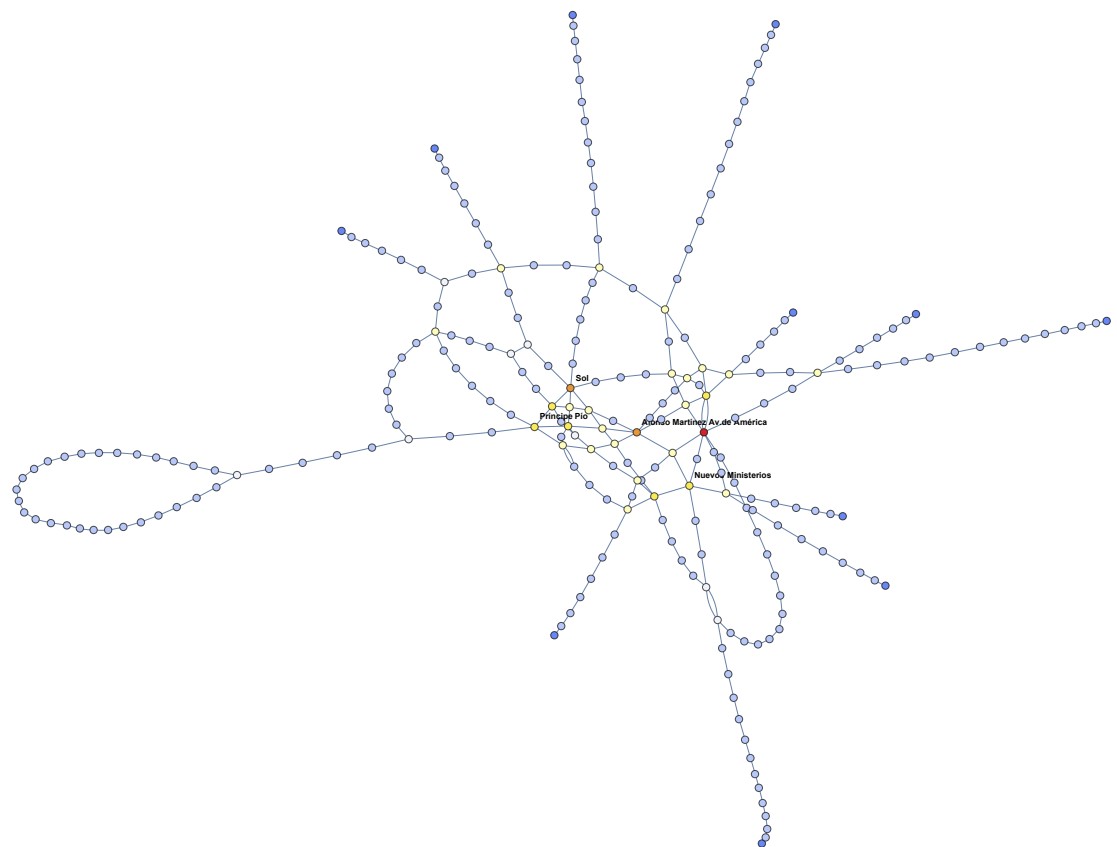

**Figure 3.** The five stations with highest degree centrality.

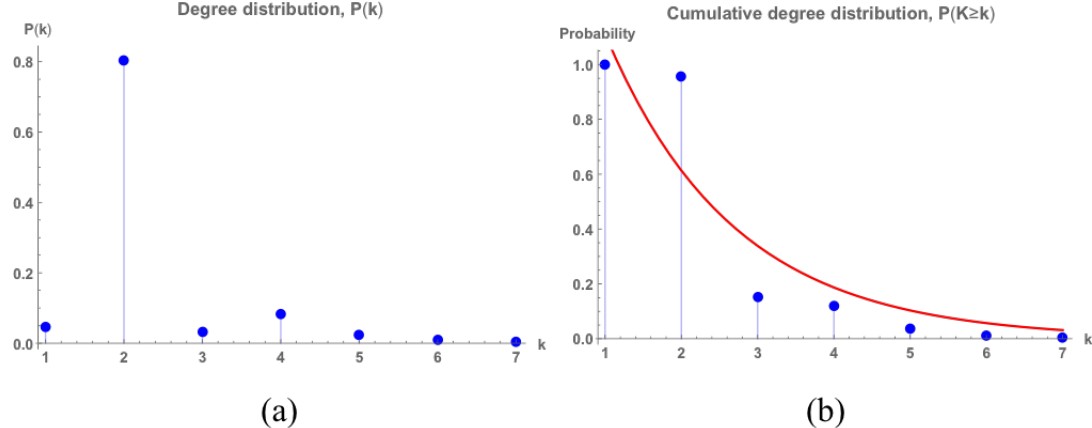

**Figure 4.** (**a**) Degree distribution of Madrid metro network; and (**b**) cumulative degree distribution of Madrid metro network.

Moreover, the diameter of the network is $D = 44$ and the average path length is $L \approx 14.6822$. In Table 3, the most central stations considering its eccentricity are shown. Note that the radius of Madrid metro network is 22 and its center is the station "Príncipe Pío", that is, this is the location that minimizes the maximum distance to any other station of the network. Obviously, its adjacent nodes (the five other stations introduced in Table 3) are the subsequent nodes with minimal eccentricity (the first five stations are illustrated in Figure 5).

**Table 3.** The six stations with the highest eccentricity.

| Station | Subway Lines | Eccentricity |
|---|---|---|
| Príncipe Pío | 6, 10, R | 22 |
| Lago | 10 | 23 |
| Puerta del Ángel | 6 | 23 |
| Plaza de España | 3, 10 | 23 |
| Argüelles | 3, 4, 6 | 23 |
| Opera | 2, 5, R | 23 |

In Table 4, the stations with the highest clustering centrality are shown. Furthermore, the five most central ones are shown in Figure 6. The most central considering these coefficient are "Callao" ($C_{CLU} \approx 0.3333$) and "Diego de León" with $C_{CLU} = 0.2$. These two stations belongs to Line 5. Moreover, the average clustering coefficient is $\widetilde{C}_{CLU} = 0.0077$.

**Table 4.** The ten stations with the highest clustering coefficient.

| Station | Subway Lines | Clustering Coefficient |
|---|---|---|
| Callao | 3, 5 | 0.3333 |
| Diego de León | 4, 5, 6 | 0.2 |
| Núñez de Balboa | 5, 9 | 0.1666 |
| Ventas | 2, 5 | 0.1666 |
| Manuel Becerra | 2, 6 | 0.1666 |
| Gran Vía | 1, 5 | 0.1666 |
| Tribunal | 1, 10 | 0.1666 |
| Bilbao | 1, 4 | 0.1666 |
| Sol | 1, 2, 3 | 0.1333 |
| Opera | 2, 5, R | 0.1 |

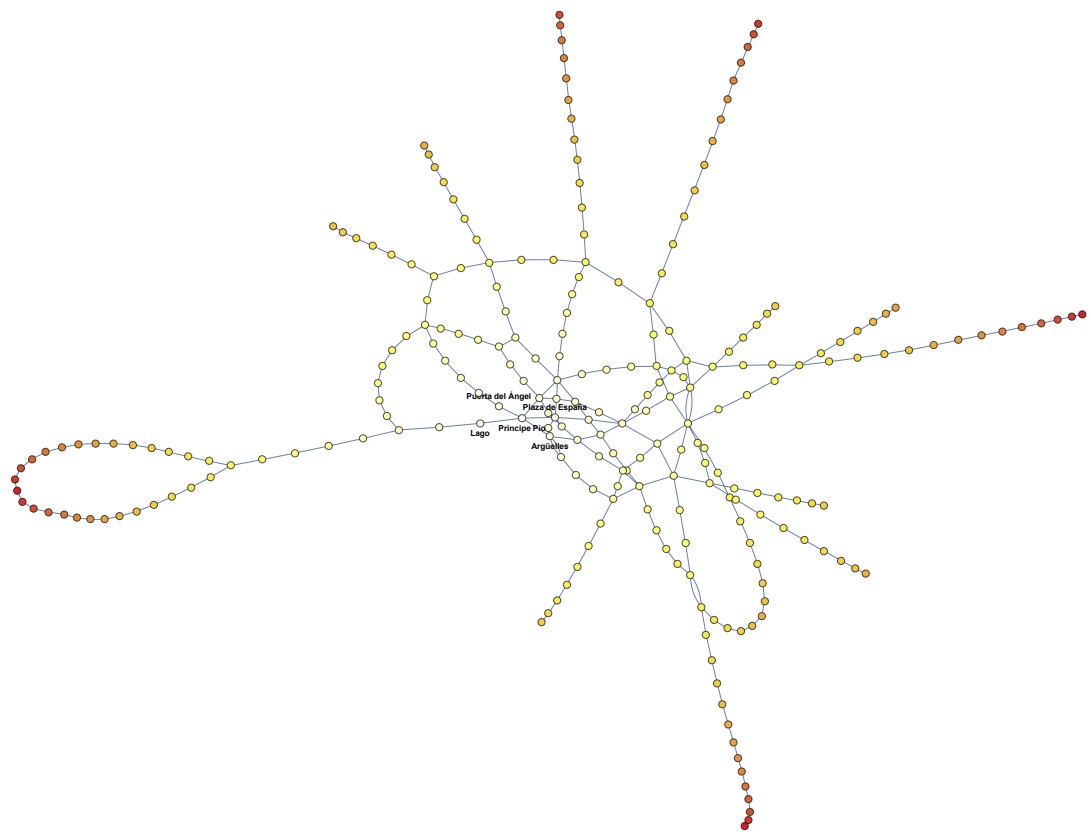

**Figure 5.** The five stations with highest eccentricity.

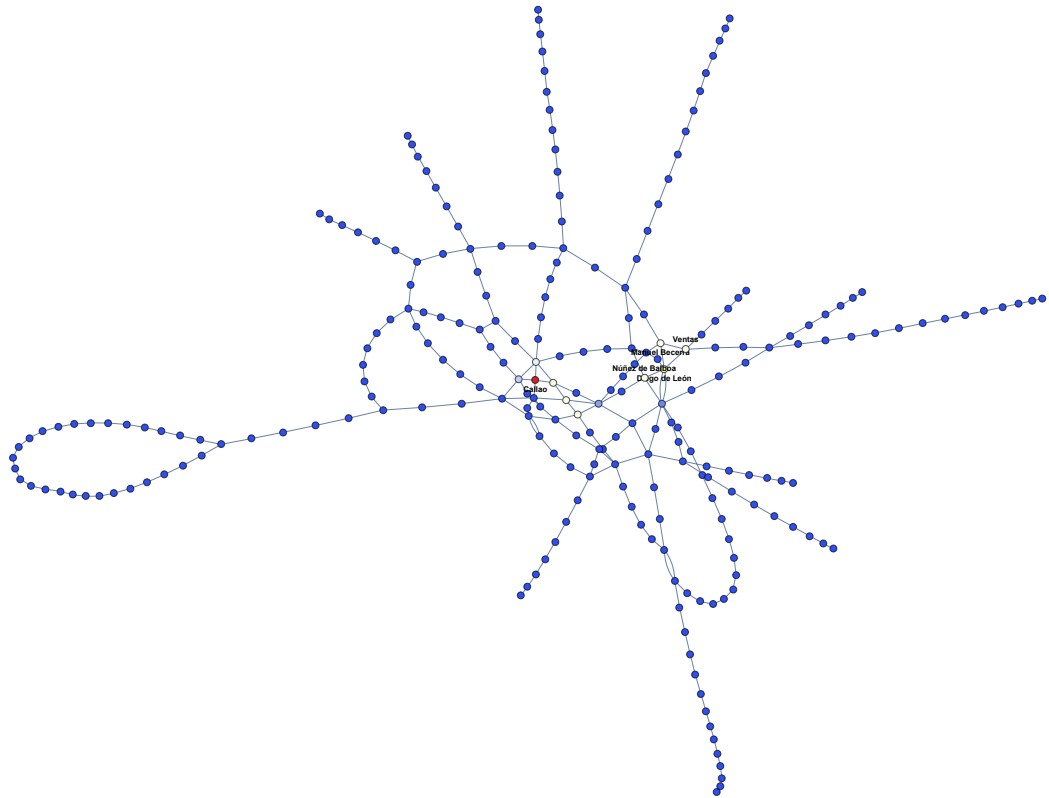

**Figure 6.** The five stations with highest clustering coefficient.

The results obtained from the computation of the closeness centrality are shown in Table 5 and illustrated in Figure 7. In our case, $0.0001481 \leq C_{CL}(v_i) \leq 0.0004575$, and $0.03585 \leq \tilde{C}_{CL}(v_i) \leq 0.1107$ for every $i$. Note that the two stations with the highest closeness centrality are "Gregorio Marañón" ($C_{CL} \approx 0.0004575$) and "Alonso Martínez" ($C_{CL} \approx 0.0004568$). Both stations belong to Line 10.

**Table 5.** The ten stations with the highest closeness centrality.

| Station | Subway Lines | Closeness Centrality | Normalized Closeness Centrality |
|---|---|---|---|
| Gregorio Marañón | 7, 10 | 0.0004575 | 0.1107 |
| Alonso Martínez | 4, 5, 10 | 0.0004568 | 0.1106 |
| Av. de América | 4, 6, 7, 9 | 0.0004466 | 0.1081 |
| Tribunal | 1, 10 | 0.0004464 | 0.1080 |
| Núñez de Balboa | 5, 9 | 0.0004395 | 0.1064 |
| Rubén Darío | 5 | 0.0004355 | 0.1054 |
| Bilbao | 1, 4 | 0.0004334 | 0.1049 |
| Plaza de España | 3, 10 | 0.0004330 | 0.1048 |
| Nuevos Ministerios | 6, 8, 10 | 0.0004325 | 0.1047 |
| Diego de León | 4, 5, 6 | 0.0004299 | 0.1040 |

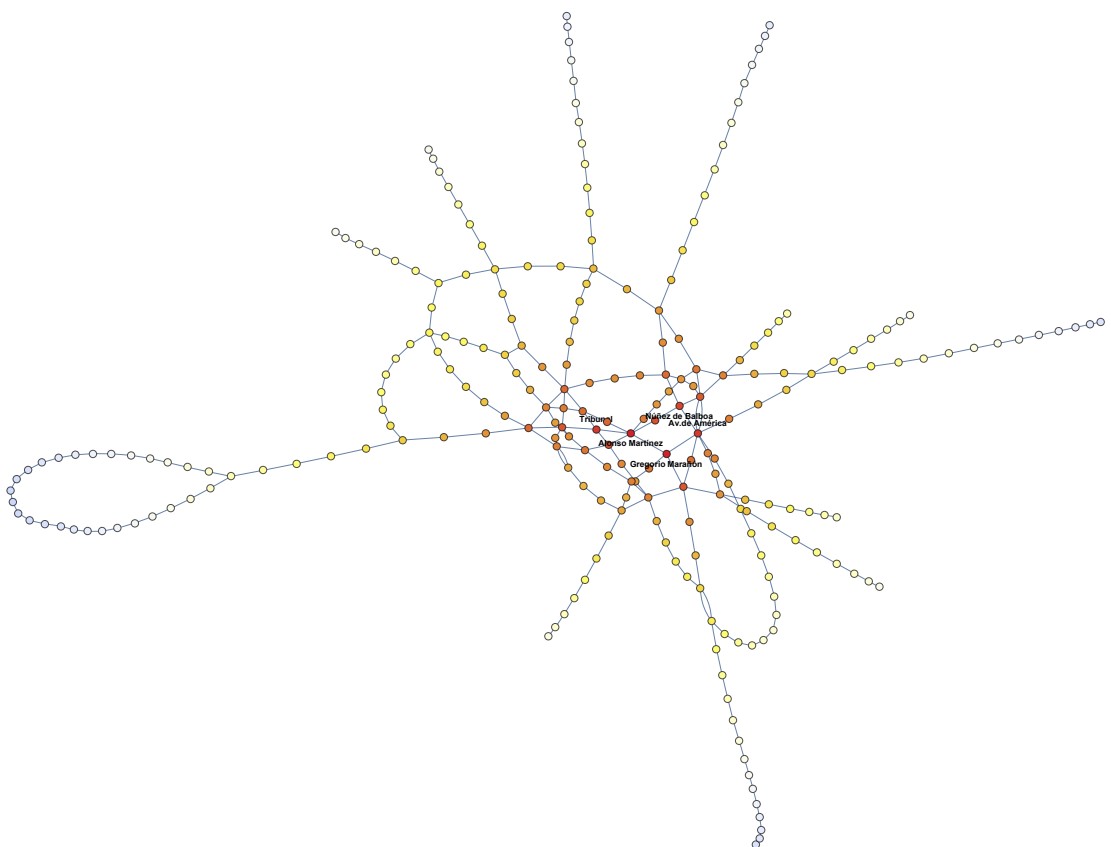

**Figure 7.** The five stations with highest closeness centrality.

Finally, the results dealing with the betweenness centrality are displayed in Table 6 and Figure 8.

**Table 6.** The ten stations with the highest betweenness centrality.

| Station | Subway Lines | Betweenness Centrality | Normalized Betweenness Centrality |
|---|---|---|---|
| Gregorio Marañón | 7, 10 | 9350 | 0.3180 |
| Príncipe Pío | 6, 10, R | 8918 | 0.3033 |
| Nuevos Ministerios | 6, 8, 10 | 8855 | 0.3012 |
| Av. de América | 4, 6, 7, 9 | 8839 | 0.3006 |
| Alonso Martínez | 4, 5, 10 | 8629 | 0.2935 |
| Casa de Campo | 5, 10 | 7406 | 0.2519 |
| Lago | 10 | 7137 | 0.2427 |
| Batán | 10 | 6978 | 0.2373 |
| Tribunal | 1, 10 | 6732 | 0.2289 |
| Colonia Jardín | 10 | 6541 | 0.2225 |

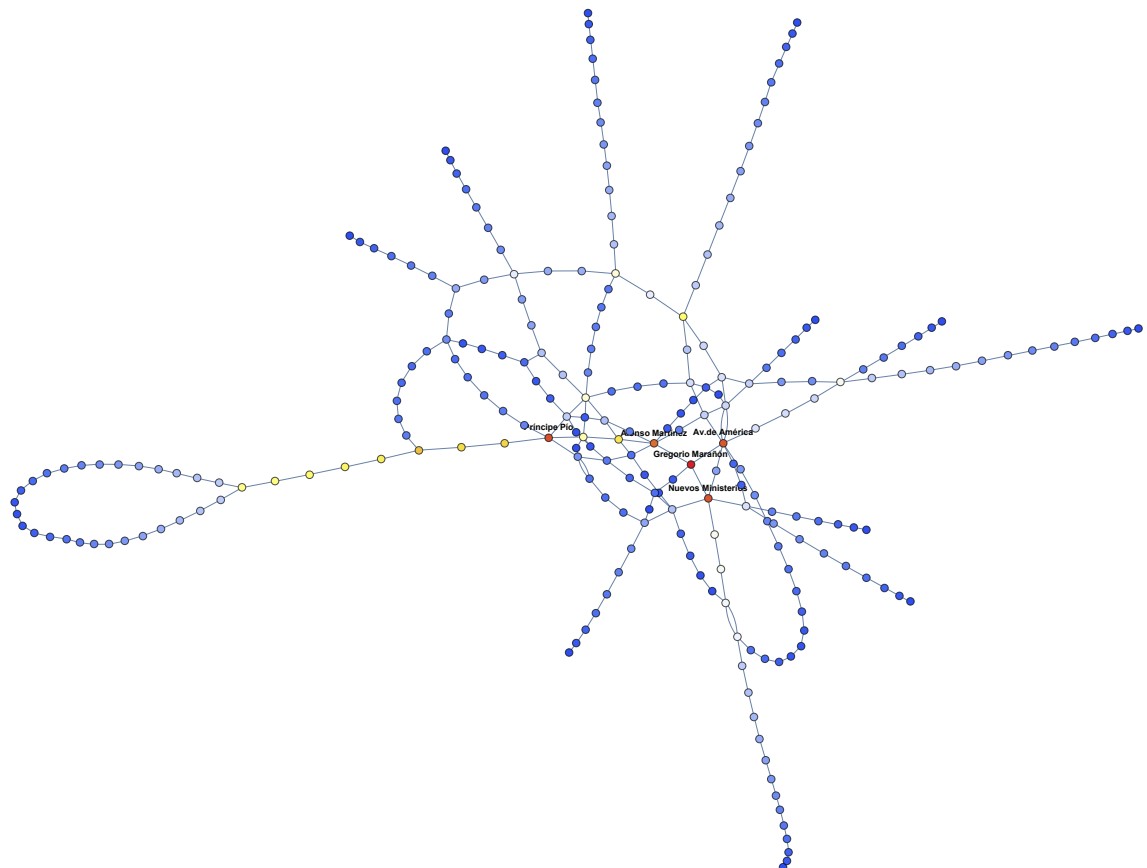

**Figure 8.** The five stations with highest betweenness centrality.

The results shown in the previous tables indicate that some stations play a central role in the structural definition of the network. For example, although the degree of "Gregorio Marañón" is not high (it belongs to two lines), this station is a very important structural piece of the subway network since it possesses the highest value of closeness and betweenness centralities. In addition, we want to highlight the role of "Avenida de América" in the structural cohesion of Madrid metro network: it belongs to four lines (its degree is the highest) and it has high coefficients in the case of closeness and betweenness centrality. Furthermore, "Nuevos Ministerios" and "Alonso Martínez" are also important centrality stations. In Figure 9, the location of these stations is illustrated: note that they are directly connected in pairs. Finally, it is also remarkable that stations with highest eccentricity are not central considering betweenness and clustering.

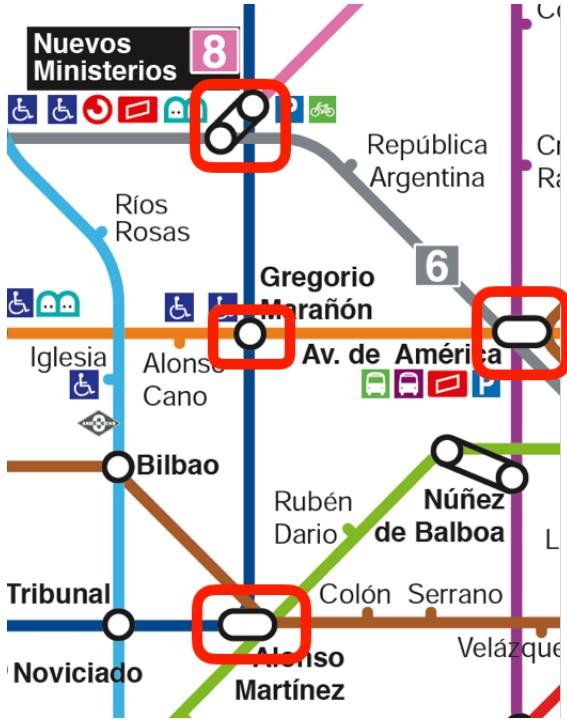

**Figure 9.** The most central stations of Madrid metro network.

### 3.3. The Role of Metro lines 5, 6 and 10

From the results presented in Tables 2, 4, 5 and 6, it was derived that Lines 5, 6 and 10 have high structural importance in the network. All these lines exhibit high values in degree, clustering, closeness and betweenness. Line 5 has six stations in the top ten of clustering (in fact, the top four belong to Line 5) and four stations in the top ten of closeness. Line 6 has five stations in the top ten of degree centrality and three stations in the top ten of betweenness. However, the most outstanding is Line 10 since nine stations of the top ten for betweenness belongs to Line 10, it has five stations in the top ten of closeness and four stations in the top ten of degree. Furthermore, the center (and two other stations with high eccentricity) belongs to Line 10. This gives us an idea of the prominence of such line in Madrid metro network.

We next studied how the (separately) removal of these lines would affect the behavior of the network. In Table 7, the most important global coefficients are shown when each of these lines is eliminated.

**Table 7.** Structural coefficients obtained after the removal of one line.

| Coefficients | Line 5 | Line 6 | Line 10 |
| --- | --- | --- | --- |
| $n$ | 221 | 229 | 222 |
| $m$ | 248 | 252 | 250 |
| $d$ | 0.01007 | 0.009615 | 0.01011 |
| $\langle k \rangle$ | 2.217 | 2.192 | 2.234 |
| Fitting $(h_1, h_2)$ | $(2.055, -0.6146)$ | $(2.065, -0.6212)$ | $(2.040, -0.6077)$ |
| $L$ | 15.12 | $\infty$ | $\infty$ |
| $D$ | 44 | $\infty$ | $\infty$ |
| Mean clustering | 0.0023 | 0.0067 | 0.0067 |

Obviously, when some nodes and edges are removed from a network, its density $d$ and average degree $\langle k \rangle$ decrease. From the new degree distributions, it is shown that the probability of obtaining nodes with degree two increases to values close to 0.5. Note that, when Line 5 is eliminated, the average path length $L$ increases, although the diameter $D$ remains constant. In this case, all stations have a null

clustering coefficient with the exception of "Alonso Martínez", "Tribunal" and "Bilbao" whose values are equal to 0.1667; the closeness centrality hardly varies; and the station with the highest betweenness centrality happens to be "Avenida de América" with 0.3632. On the other hand, when Line 6 or Line 10 is removed, the resulting network is disconnected. In addition, the mean clustering coefficient becomes smaller than the original one but the top ten does not vary much. Finally, the normalized betweenness coefficient of "Gregorio Marañón" when Line 6 is removed is remarkable: it increases from 0.3180 to 0.4269. In this case, when Line 10 is removed and "Gregorio Marañón" only belongs to Line 7, its normalized betweenness centrality drops to 0.0213, being "Avenida de América" the node with highest betweenness (0.2540).

### 3.4. Analysis of the Central Core

Line 6 is a circular subway line, which, in some way, delimits the extended central urban area of the city (see Figure 10). Therefore, it seems to be interesting to study this subnetwork in order to identify the principal differences with respect to the original one.

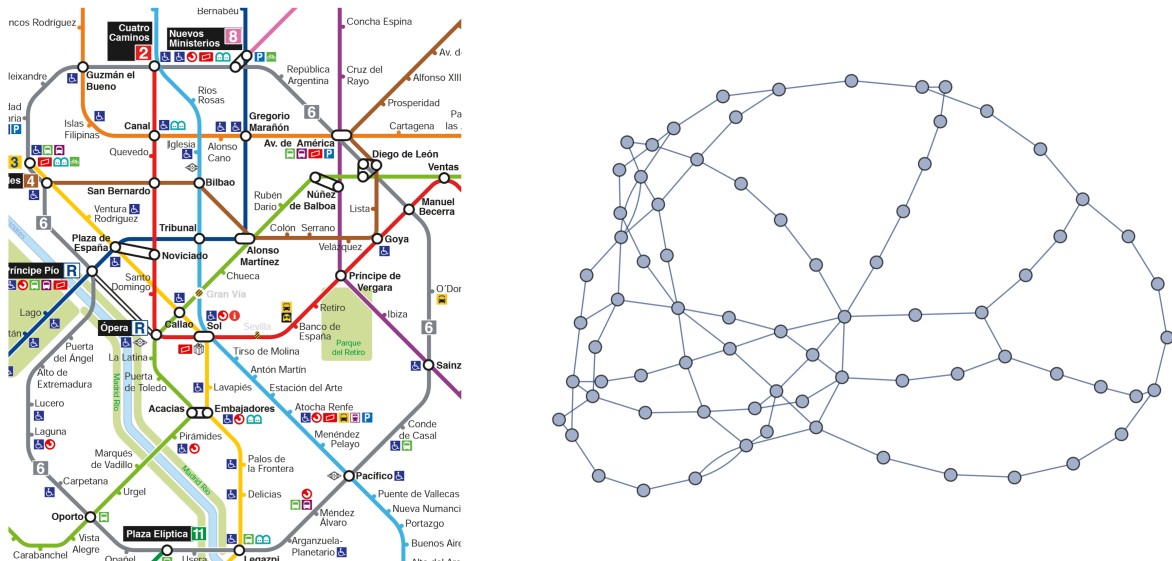

**Figure 10.** Central core (delimited by Line 6) of Madrid metro network.

In this case, $n = 74$ and $m = 101$, thus the density is $d \approx 0.03665$, which is significantly larger than the original one (0.009421). This is an expected fact since we have removed the "tentacles" (outside of Line 6) of the network, which are constituted by several stations with degree 2 and all stations of degree 1. Obviously, the diameter and the average path length drop from 44 and 14.68 to 13 and 5.948, respectively. In Figure 11a,b, the degree and cumulative degree distributions are shown, respectively. The fitting function of the cumulative degree distribution is $h(x) = 1.8368e^{-0.4834x}$.

The mean clustering coefficient increases from 0.0077 to 0.0218. Now, the most central station is "Sol" (Lines 1, 2 and 3); it is the station with highest closeness centrality with 0.2320 ("Gregorio Marañón" goes down from the first place to the thirteenth) followed by "Tribunal" (0.2277) and "Plaza de España" (0.2222), and it is also the station with highest betweenness centrality: 0.3046 ("Gregorio Marañón" also goes down to the ninth place).

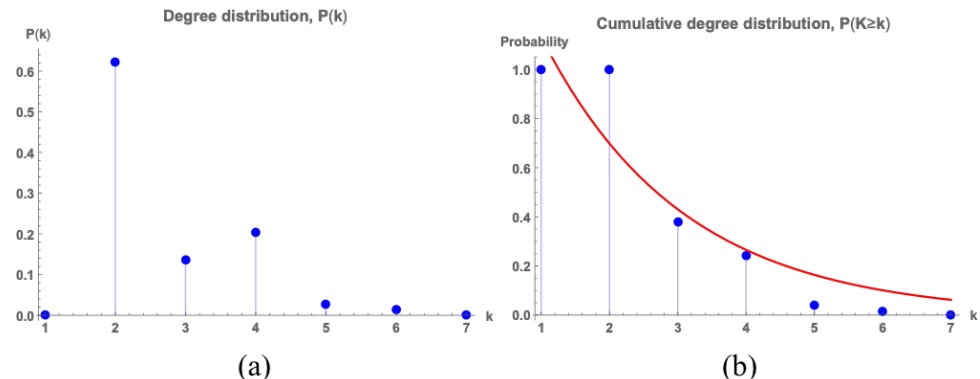

**Figure 11.** (**a**) Degree distribution of central core; and (**b**) cumulative degree distribution of central core.

## 4. Robustness Analysis of Madrid Subway Network

### 4.1. Analysis of Robustness by Removing Stations

Failures of subway networks can have enormous impact on our society, thus the analysis of the robustness is very important when studying subway networks. The robustness of networks reflects the extent to which the networks can solve possible (intentional or unintentional) failures by offering alternative routes that overcome the attacked edges or nodes [41].

The robustness metrics described in Section 2.2 were computed for the Madrid subway network and briefly analyzed. Then, the critical thresholds under random failures and targeted attacks were obtained through simulations.

Table 8 shows the values of the theoretical and numerical robustness metrics computed.

**Table 8.** Robustness metrics in Madrid subway networks.

| Coefficients | Madrid Subway Network |
|---|---|
| Nodes, $N$ | 243 |
| Edges, $M$ | 280 |
| Assortativity coefficient | 0.2963 |
| Robustness indicator, $r^T$ | 0.1440 |
| Effective graph conductance, $C_G$ | 0.0008631 |
| Average efficiency, $E_G$ | 0.1053 |
| Average clustering coefficient, $C_{CG}$ | 0.007741 |
| Algebraic connectivity, $\lambda_{N-1}$ | 0.003767 |
| Normalized algebraic connectivity, $\widetilde{\lambda}_{N-1}$ | 0.00001550 |
| Natural connectivity, $\lambda$ | 1.0489 |
| Normalized natural connectivity, $\widetilde{\lambda}$ | 0.004416 |
| Degree diversity, $\kappa$ | 2.693 |
| Percolation limit, $p_c$ | 0.4093 |
| Critical threshold $f_{90\%}$-degree | 0.02880 |
| Critical threshold $f_{90\%}$-betweenness | 0.00823 |
| Critical threshold $f_{90\%}$-random | 0.03292 |
| Critical threshold $f_c$-degree | 0.51028 |
| Critical threshold $f_c$-betweenness | 0.99588 |
| Critical threshold $f_c$-random | 0.92181 |

Madrid metro network is slightly degree assortative ($r \approx 0.2963$), which suggests that a significant fraction of stations with a low degree connect to other low degree stations. Note that, if we remove Line R (which only has one connection between the stations "Opera"—L2 and L5—and "Príncipe Pío"—L6 and L10), the assortativity coefficient decreases to 0.2810. Furthermore, if Line L5 or L10 is removed from the network, the respective assortative coefficient is $r \approx 0.1757$ or $r \approx 0.2191$, respectively. In addition, if the most central stations ("Avenida de América", "Gregorio Marañón",

and "Alonso Martínez") are removed, the assortativity of the network decreases to 0.2953, 0.2789, and 0.2658, respectively.

The robustness indicator of Madrid subway network is $\bar{r}^T = 0.1440$ since $M^m = 3$. This indicator favors the metro networks which have many alternative paths between any pairs of nodes and disadvantages those metro networks which have a large number of nodes but few alternative paths.

The effective graph conductance is $C_G \approx 0.0008631$, thus, according to this value, Madrid subway network is not robust. This is due to this measure accounts not only the number of alternative paths but also the length of each alternative path, hence it favors the star-like topology with a small average shortest path length.

The average efficiency is $E_G \approx 0.1053$, thus the global connectivity is quite poor. It is important to note that the maximum value is obtained in fully connected networks and it is equal to 1.

The average clustering coefficient of Madrid subway network ($\tilde{C}_{CLU} \approx 0.007741$) is lower than that of the subways of some other European cities such as Copenhagen ($\tilde{C}_{CLU} = 0.0769$), London ($\tilde{C}_{CLU} = 0.0387$ or Paris ($\tilde{C}_{CLU} = 0.0157$) [24]. It is important to note that this coefficient measures the global connectivity of the network as it assesses to what extent the neighbors of a node are connected to another one.

In our case, both vertex connectivity $\kappa_V$ and edge connectivity $\kappa_E$ are equal to 1 since there exist monotonic nodes, and consequently $0 \leq \lambda_{N-1} \approx 0.003 \leq 1$. The higher this value is, the more difficult it is to disintegrate the network, thus it is quite easy to disintegrate Madrid network into components.

The natural connectivity uses the number of alternative paths to quantify the robustness of a network. The normalized natural connectivity $\widetilde{\lambda}$ in our case is approximately equal to 0.004, therefore, since it takes values between 0 and 1, it seems that Madrid subway is not very robust.

The degree diversity of Madrid metro network is $\kappa \approx 2.693$. Note that for a network to have a giant component most nodes that connect to it must be connected to at least two other vertices. This leads to the Molloy–Reed criterion that states that a randomly connected network has a giant component if $\kappa > 2$ [42]. In our case, this condition is met with difficulty; indeed, the percolation limit is $p_c \approx 0.4093$, which is similar to this coefficient exhibited by the largest connected component of random graph networks defined by the Erdös–Renyi algorithm with edge probability $p \approx 0.006897$. This value is smaller than that presented by other public transport networks [43].

Figure 12 describes the performance changes of the largest connected component of Madrid subway subjected to different network failures. It shows that the most serious network attack is the highest betweenness based attacks, the second most serious attack is the largest degree-node attacks and finally random disruption of stations results in minimum damage to the network. In this way, if the ten stations with the greatest betweenness centrality are eliminated, the size of the largest connected component is reduced by 80%. When stations are eliminated according the degree, the elimination of ten stations can result in more than 65% reduction in the size of the largest connected component (Figure 13). In the case of random attacks, ten stations removed only results in 8% of reduction in the size of the largest connected component. Thus, Madrid subway network is fragile when subjected to intentional attacks, and it is quite robust against random attacks. The lowest value of the critical thresholds $f_{90\%}$ was obtained when nodes are removed according to their betweenness centrality (Table 8). On the other hand, we have that the critical thresholds $f_c$ of the fraction of removed nodes of the network subjected to largest degree-node based attacks is the smallest one.

Figure 14 depicts the changes in the network efficiency subjected to different network failures. It shows that the highest betweenness based attacks will result in the highest damage and the random attacks cause the minimum damage among these different attacks. Thus, if the ten stations with the greatest betweenness centrality are eliminated, the network efficiency is reduced more than 70%. The damage caused by largest degree attacks is slightly smaller than the damage caused by highest betweenness based attacks.

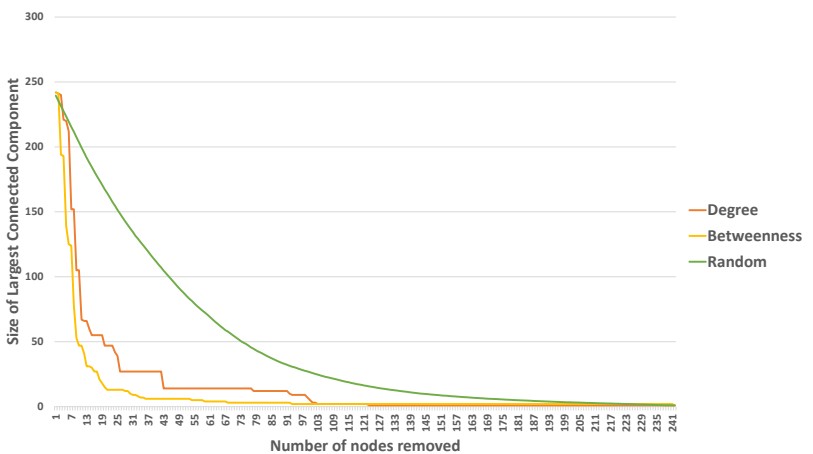

**Figure 12.** The changes of the size of largest connected component with different attack protocols.

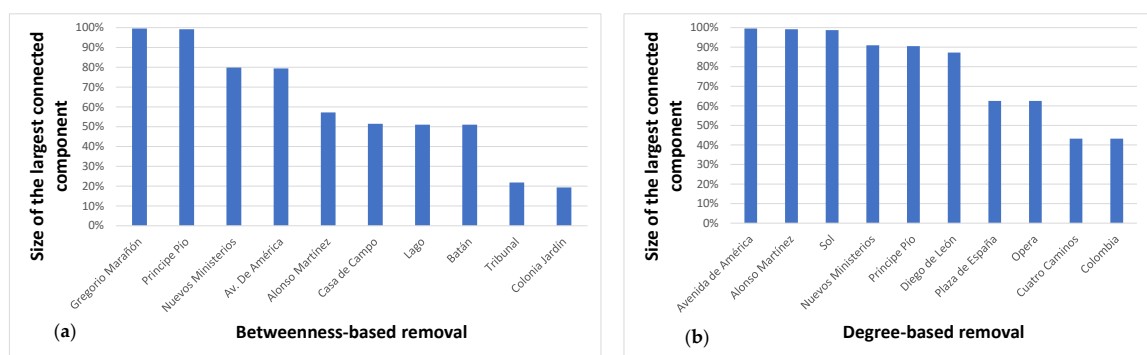

**Figure 13.** (**a**) Size of the largest connected component when each station is removed (10 stations are selected based on betweenness); and (**b**) size of the largest connected component when each station is removed (10 stations are selected based on degree).

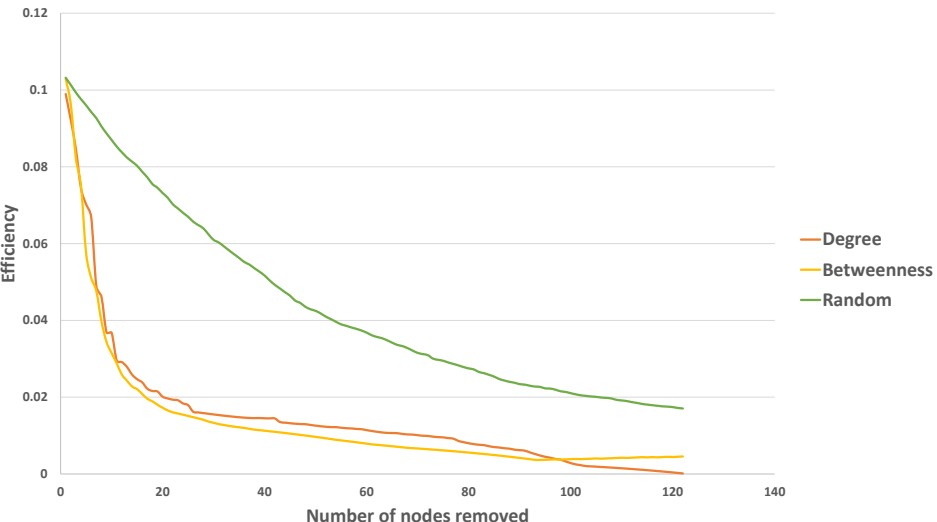

**Figure 14.** The changes of the network efficiencies with different attack protocols.

The changes in the average betweenness for nodes subjected to different network failures are shown in Figure 15. Again, it can be seen how Madrid subway is quite robust against random attacks but is fragile when it is subjected to intentional attacks (largest degree node-based attacks and highest betweenness node-based attacks). Furthermore, it can be observed that the highest betweenness node-based attacks will result in the most serious damage to the network. When one

node/station is removed, the resulting network could have the same number of connected components or not (the number of connected components increases). In the first case, if the removed node has high centrality values (degree and/or betweenness), the average betweenness of the network usually increases (since the average number of the paths between any pairs of nodes decreases significantly)—this is exactly what happens in the beginning. On the other hand, if the node removed has similar centrality coefficients than the remaining nodes in the network, the average betweenness is slightly decreased. In the second case, when more connected components appear, the average betweenness decreases. This behavior is similar to that exhibited by, for example, Shanghai [44].

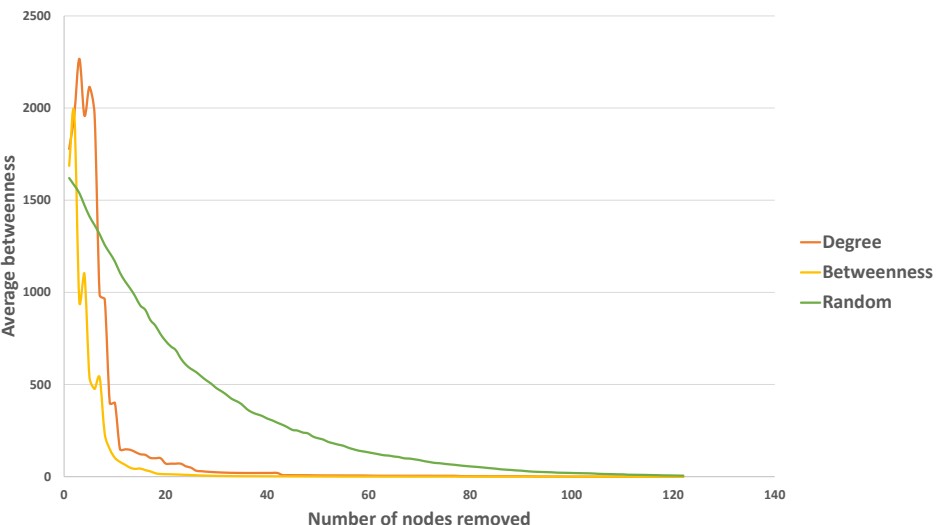

**Figure 15.** The changes of the average betweenness with different attack protocols.

Figure 16 shows the changes of the number of connected components when the network is subjected to the three different networks failures. According to Figure 15, with the increase of the number of the removed nodes, the number of connected components increases with the quickest velocity when the largest degree node-based or the highest betweenness attack protocol are applied to the network. In this sense, when the three stations with the highest betweenness are eliminated, the network disintegrates into three connected networks, and, when the number of stations eliminated is ten, the number of connected components is eleven.

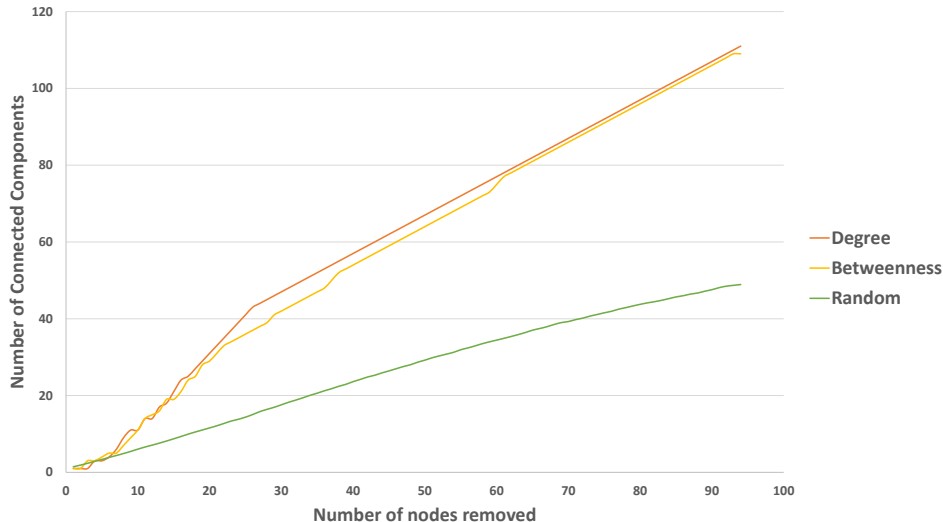

**Figure 16.** The changes of the number of connected components with different attack protocols.

With all this, it is shown that malicious attacks can effectively destroy the network.

### 4.2. Analysis of Robustness by Removing Lines

In this section, an analysis of the robustness of the network is presented when metro lines were removed instead of nodes/stations. For the sake of simplicity, we only studied those cases when one or two metro lines were removed (it does not seem realistic to consider eliminating three or more lines).

First, if only one line is removed, then 12 possible metro networks are obtained. In Tables 9 and 10, the numeric values of the most important robustness coefficients are shown when each line is removed. The main conclusion for these empirical results is that removing lines 2, 6 or 10 affects the reliability of the metro network more than removing the other lines.

**Table 9.** Robustness metrics when only one line is removed.

| Removed Line | Global Efficiency | Algebraic Connectivity | Average Degree |
|:---:|:---:|:---:|:---:|
| 12 | 0.1162 | $41.10 \times 10^{-6}$ | 0.01068 |
| 11 | 0.1031 | $9.331 \times 10^{-6}$ | 0.009618 |
| 10 | 0.09369 | 0 | 0.009987 |
| 9 | 0.1044 | $10.25 \times 10^{-6}$ | 0.01035 |
| 8 | 0.1012 | $9.346 \times 10^{-6}$ | 0.009583 |
| 7 | 0.1047 | $10.31 \times 10^{-6}$ | 0.01040 |
| 6 | 0.09166 | 0 | 0.009500 |
| 5 | 0.1011 | $9.482 \times 10^{-6}$ | 0.01004 |
| 4 | 0.1005 | $9.645 \times 10^{-6}$ | 0.009712 |
| 3 | 0.1019 | $9.563 \times 10^{-6}$ | 0.009712 |
| 2 | 0.09920 | $9.486 \times 10^{-6}$ | 0.009554 |
| 1 | 0.1022 | $10.19 \times 10^{-6}$ | 0.0102 |

**Table 10.** Robustness metrics when only one line is removed.

| Removed Line | Natural Connectivity | Percolation Limit |
|:---:|:---:|:---:|
| 12 | 0.004992 | 0.4137 |
| 11 | 0.004462 | 0.3928 |
| 10 | 0.004586 | 0.3552 |
| 9 | 0.004745 | 0.3747 |
| 8 | 0.004443 | 0.3853 |
| 7 | 0.004809 | 0.3756 |
| 6 | 0.004228 | 0.3014 |
| 5 | 0.004535 | 0.3476 |
| 4 | 0.004389 | 0.3411 |
| 3 | 0.004438 | 0.3661 |
| 2 | 0.004383 | 0.3469 |
| 1 | 0.004663 | 0.3605 |

For example, when global efficiency is considered, the elimination of any of these lines causes the global efficiency to drop by 12%. The case of algebraic connectivity is paradigmatic: when Lines 6 and 10 disappear from the whole network, then it stops being connected. In addition, if Line 2 or 5 is removed, the metro network remains connected but its robustness greatly decreases.

It should be notice the importance of Line 2 to the structural cohesion of the network. When this line is removed, the numeric values of natural connectivity and percolation limit decrease considerably. Nevertheless, these computations reveals that the most central line when robustness is tackled is Line 6; this is a circular line with a great number of stations (note that the normalized average degree is about 0.0095—the minimum when one line disappears) and with a high connectivity to the rest of lines. Consequently, it seems reasonable that great efforts be made in the maintenance of this metro line.

Finally, the previous results can be corroborated when two lines are removed from the Madrid subway network. For example, if we compute the global efficiency in this new situation, we can see

the central role of Lines 6 and 10. In Figure 17, a 3D point cloud is represented from data $(i, j, E_{ij})$ where coordinates $i$ and $j$ stand for the removing lines ($1 \leq i, j \leq 12$) and $0.08 \leq E_{ij} \leq 0.14$ is the global efficiency of the metro network when lines $i$ and $j$ are removed. Note that, when Lines 6 and 10 ($i, j = 6, 10$) disappear, the fitting surface exhibits a valley.

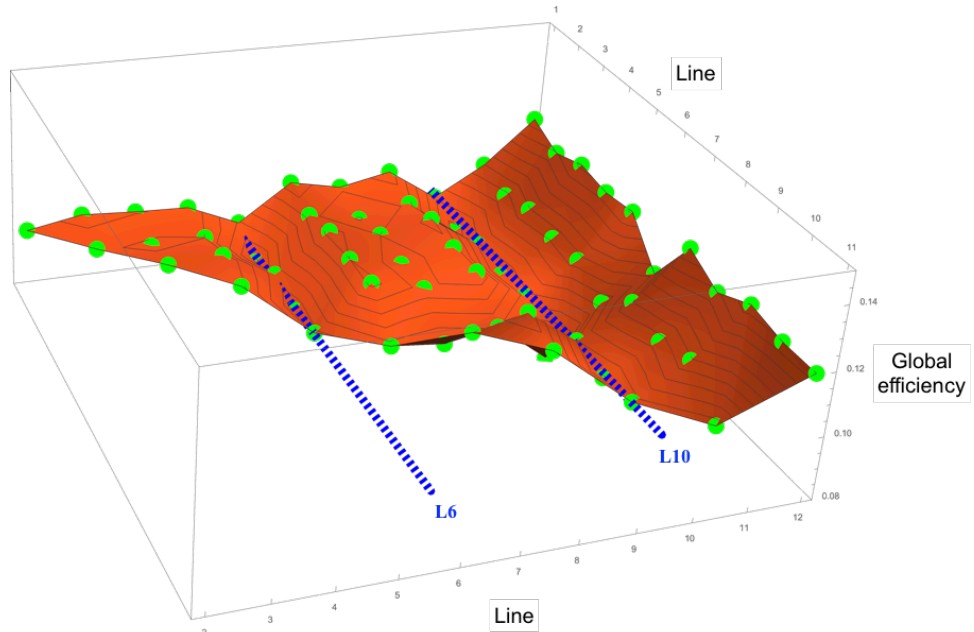

**Figure 17.** Global efficiency when two lines are removed.

## 5. Conclusions and Future Work

In this work, a detailed analysis of Madrid metro network was done following the paradigm of Complex Network Analysis. Specifically, the most important centrality measures and coefficients were computed not only for the whole network but also for reduced networks obtained by removing sensitive lines and stations. Moreover, the most important robustness coefficients were also computed for the subway network.

In this sense, it was shown that the most central stations are "Gregorio Marañón", "Alonso Martínez" and "Avenida de América", and the most central subway lines are Line 5 ("Alameda de Osuna"–"Casa de Campo"), Line 6 (circular line from "Laguna"–"Lucero") and Line 10 ("Hospital Infanta Sofía"–"Puerta del Sur").

Taking into account the results derived from the study of the robustness of Madrid metro network, we can state that Madrid metro network is more vulnerable to attacks than other public transport networks. In this sense, the stations "Gregorio Marañón" and "Avenida de América" play an important role in ensuring the robustness of the transportation network.

Further work will analyze in a detailed way the Madrid metro network considering different topological representations (P-space, C-space, etc.), additional transport lines (light rail network), sociological and geographical coefficients. On the other hand, this work exhibits some limitations in relation to its practical application. Specifically, it would also be interesting to consider in the future robustness studies additional information such as the number and periodicity of trains, the number of passengers that use the different stations and lines, etc. This would provide us an analysis of the number of passengers directly or indirectly damaged by the failure of a single node, or by the number of passengers not influenced by the failure.

**Author Contributions:** E.F.B. and A.M.d.R. conceived and designed the study; A.M.d.R. performed the computational implementation; E.F.B. provided the interpretation of results; and the paper was written, edited and revised by both authors.

**Funding:** This research was funded by Ministerio de Ciencia, Innovación y Universidades (MCIU, Spain), Agencia Estatal de Investigación (AEI, Spain), and Fondo Europeo de Desarrollo Regional (FEDER, UE) under project with reference TIN2017-84844-C2-2-R (MAGERAN) and the project with reference SA054G18 supported by Consejería de Educación (Junta de Castilla y León, Spain).

**Acknowledgments:** The authors thank the anonymous reviewers for their valuable suggestions and comments that have greatly improved this work.

**Conflicts of Interest:** The authors declare no conflict of interest. The funders had no role in the design of the study; in the collection, analyses, or interpretation of data; in the writing of the manuscript, or in the decision to publish the results.

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
