# Peer review of "Study of the Structural and Robustness Characteristics of Madrid Metro Network"

_sustainability, doi:10.3390/su11123486_

Reviewer 1 Report

This manuscript investigated the structural and robustness characteristics of Madrid metro network using network analysis techniques. This is one of the hottest topics about network analysis and its applications of the real world in recent years. However, there are several limitations in the manuscript should be completely revised before it can be accepted for publication.

1. Section I: 
1.1 The authors should clearly declare the motivation of the study, why Madrid metro network needs detailed study of the structural and robustness characteristics using network analysis techniques. 
1.2 The authors also should explain the contributions of this study to the Madrid metro. Only analysis of Madrid metro as a complex network using network centrality and other metrics is insufficient. 
1.3 The third paragraph is unnecessary. Maybe it could be one of the motivations of this study. The authors should improve the contextual relationships.

2. Section II:
2.1 In this section, the authors have sufficiently reviewed network centralities and other metrics. I recommend that the authors separate these metrics into different classes by their characteristics, which would be more clearly for readers to understand network topology. For example, degree is a local centrality measure, and betweenness is a global centrality measure. 
2.2 I recommend that the authors improve Section II with several real-world analyses and applications, which have already applied network analysis techniques, such as power grid, Internet and supply chain. I also recommend the authors to add the references as follows.
Albert, R.; Jeong, H.; Barabasi, A.L. Error and attack tolerance of complex networks. Nature 2000, 406,378–382.

Barabasi, A.L.; Bonabeau, E. Scale-free networks. Sci. Am. 2003, 288, 60–69.

Guimera, R.; Amaral, L.A.N. Functional cartography of complex metabolic networks. Nature 2005, 433,895–900.

Demetrius, L.; Manke, T. Robustness and network evolution-an entropic principle. Physica A 2005, 346, 682–696.

J.-P. Onnela et. al. Structure and tie strengths in mobile communication networks. PNAS, 2007, 104(18), 7332-7336.

Kajikawa, Y.; Takeda, Y.; Sakata, I.; Matsushima, K. Multiscale analysis of interfirm networks in regional clusters. Technovation 2010, 30, 168–180.

Kim, Y.; Choi, T.Y.; Yan, T.; Dooley, K. Structural inestigation of supply networks: A social network analysis approach. J. Oper. Manag. 2011, 29, 194–211.

Zuo, Y.; Kajikawa, Y. Toward a Theory of Industrial Supply Networks:
A Multi-Level Perspective via Network Analysis. Entropy, 2017, 19, 382.

3. Section III and IV:
3.1 I recommend the author to color and scale Figure 2 by different lines and classes (monotonic nodes, transfer nodes and termini nodes), which would be more intuitive for reader. 
3.2 The experimental results are almost shown by tables. I recommend that such results would be better to present in a colored and scaled network. 
3.3 However, this study sufficiently analyzed the topology of Madrid metro network. The author neglected passengers flow and its demand for different lines. 
3.4 I wonder that robustness analysis of Madrid metro network is the most important part in this study. However, the use cases are oversimple. Network evolution such as non-contact islands in the subway system should be considered. 
3.5 I believe that 3.3 and 3.4 constitute a useful extension of the research activity.

4. Section V: 
The authors should reveal the limitations of this study, and also the plans to solve these limitations, which could be the subject of a subsequent article.

Author Response

First of all, we would like to thank you for your valuable suggestions and comments. In what follows we will explain the changes done in the manuscript to address these suggestions:

1. Section I:

1.1.The authors should clearly declare the motivation of the study, why Madrid metro network needs detailed study of the structural and robustness characteristics using network analysis techniques.

The third paragraph of the original version of the manuscript (which is the fourth paragraph of the new version of the paper) has been modified including the motivation requested.

 1.2 The authors also should explain the contributions of this study to the Madrid metro. Only analysis of Madrid metro as a complex network using network centrality and other metrics is insufficient. 

The motivation of this work has been clarified in the new version of the manuscript. Specifically, the penultimate paragraph has been modified in order to take into account this suggestion.

1.3 The third paragraph is unnecessary. Maybe it could be one of the motivations of this study. The authors should improve the contextual relationships.

The third paragraph of the original version of the manuscript (which is the fourth paragraph of the new version of the paper) has been modified including the motivation requested.

2. Section II:

2.1 In this section, the authors have sufficiently reviewed network centralities and other metrics. I recommend that the authors separate these metrics into different classes by their characteristics, which would be more clearly for readers to understand network topology. For example, degree is a local centrality measure, and betweenness is a global centrality measure. 

Yes, we agree with the reviewer. In the modified version of the paper subsection 2.1 has been divided into two new subsections: in the first one (2.2.1) the local measures and coefficients are introduced, whereas the second one (2.2.2) is devoted to global coefficients.

2.2 I recommend that the authors improve Section II with several real-world analyses and applications, which have already applied network analysis techniques, such as power grid, Internet and supply chain. I also recommend the authors to add the references as follows.

Albert, R.; Jeong, H.; Barabasi, A.L. Error and attack tolerance of complex networks. Nature 2000, 406,378–382.

Barabasi, A.L.; Bonabeau, E. Scale-free networks. Sci. Am. 2003, 288, 60–69.

Guimera, R.; Amaral, L.A.N. Functional cartography of complex metabolic networks. Nature 2005, 433,895–900.

Demetrius, L.; Manke, T. Robustness and network evolution-an entropic principle. Physica A 2005, 346, 682–696.

J.-P. Onnela et. al. Structure and tie strengths in mobile communication networks. PNAS, 2007, 104(18), 7332-7336.

Kajikawa, Y.; Takeda, Y.; Sakata, I.; Matsushima, K. Multiscale analysis of interfirm networks in regional clusters. Technovation 2010, 30, 168–180.

Kim, Y.; Choi, T.Y.; Yan, T.; Dooley, K. Structural inestigation of supply networks: A social network analysis approach. J. Oper. Manag. 2011, 29, 194–211.

Zuo, Y.; Kajikawa, Y. Toward a Theory of Industrial Supply Networks:
A Multi-Level Perspective via Network Analysis. Entropy, 2017, 19, 382.

Thank you very much for your suggestion. In the new version of the manuscript the new subsection 2.3 has appeared including a brie review about the applications of Complex Network Analysis in different areas. Moreover, the suggested references (and others) have been included.

3.3 However, this study sufficiently analyzed the topology of Madrid metro network. The author neglected passengers flow and its demand for different lines. 

Yes, we agree with the reviewer. This suggestion has been added as a future work in the section “Conclusions and future work”.

3.4 I wonder that robustness analysis of Madrid metro network is the most important part in this study. However, the use cases are oversimple. Network evolution such as non-contact islands in the subway system should be considered. 

In the new version of the manuscript a more detailed study about the robustness is included. Specifically, the number of connected components of the remaining networks (after removing nodes following different attack protocols) are computed, and also the most important robustness coefficients are computed and analyzed when metro lines disappear from the whole network.

3.5 I believe that 3.3 and 3.4 constitute a useful extension of the research activity.

Yes, we agree. This has been reflected in some way in the future work.

4. Section V:

The authors should reveal the limitations of this study, and also the plans to solve these limitations, which could be the subject of a subsequent article.

The section devoted to conclusions and future work has been modified according these suggestions.

Response to Reviewer 1 Comments

First of all, we would like to thank you for your valuable suggestions and comments. In what follows we will explain the changes done in the manuscript to address these suggestions:

1. Section I:

1.1.The authors should clearly declare the motivation of the study, why Madrid metro network needs detailed study of the structural and robustness characteristics using network analysis techniques.

The third paragraph of the original version of the manuscript (which is the fourth paragraph of the new version of the paper) has been modified including the motivation requested.

 1.2 The authors also should explain the contributions of this study to the Madrid metro. Only analysis of Madrid metro as a complex network using network centrality and other metrics is insufficient. 

The motivation of this work has been clarified in the new version of the manuscript. Specifically, the penultimate paragraph has been modified in order to take into account this suggestion.

1.3 The third paragraph is unnecessary. Maybe it could be one of the motivations of this study. The authors should improve the contextual relationships.

The third paragraph of the original version of the manuscript (which is the fourth paragraph of the new version of the paper) has been modified including the motivation requested.

2. Section II:

2.1 In this section, the authors have sufficiently reviewed network centralities and other metrics. I recommend that the authors separate these metrics into different classes by their characteristics, which would be more clearly for readers to understand network topology. For example, degree is a local centrality measure, and betweenness is a global centrality measure. 

Yes, we agree with the reviewer. In the modified version of the paper subsection 2.1 has been divided into two new subsections: in the first one (2.2.1) the local measures and coefficients are introduced, whereas the second one (2.2.2) is devoted to global coefficients.

2.2 I recommend that the authors improve Section II with several real-world analyses and applications, which have already applied network analysis techniques, such as power grid, Internet and supply chain. I also recommend the authors to add the references as follows.

Albert, R.; Jeong, H.; Barabasi, A.L. Error and attack tolerance of complex networks. Nature 2000, 406,378–382.

Barabasi, A.L.; Bonabeau, E. Scale-free networks. Sci. Am. 2003, 288, 60–69.

Guimera, R.; Amaral, L.A.N. Functional cartography of complex metabolic networks. Nature 2005, 433,895–900.

Demetrius, L.; Manke, T. Robustness and network evolution-an entropic principle. Physica A 2005, 346, 682–696.

J.-P. Onnela et. al. Structure and tie strengths in mobile communication networks. PNAS, 2007, 104(18), 7332-7336.

Kajikawa, Y.; Takeda, Y.; Sakata, I.; Matsushima, K. Multiscale analysis of interfirm networks in regional clusters. Technovation 2010, 30, 168–180.

Kim, Y.; Choi, T.Y.; Yan, T.; Dooley, K. Structural inestigation of supply networks: A social network analysis approach. J. Oper. Manag. 2011, 29, 194–211.

Zuo, Y.; Kajikawa, Y. Toward a Theory of Industrial Supply Networks:
A Multi-Level Perspective via Network Analysis. Entropy, 2017, 19, 382.

Thank you very much for your suggestion. In the new version of the manuscript the new subsection 2.3 has appeared including a brie review about the applications of Complex Network Analysis in different areas. Moreover, the suggested references (and others) have been included.

3.3 However, this study sufficiently analyzed the topology of Madrid metro network. The author neglected passengers flow and its demand for different lines. 

Yes, we agree with the reviewer. This suggestion has been added as a future work in the section “Conclusions and future work”.

3.4 I wonder that robustness analysis of Madrid metro network is the most important part in this study. However, the use cases are oversimple. Network evolution such as non-contact islands in the subway system should be considered. 

In the new version of the manuscript a more detailed study about the robustness is included. Specifically, the number of connected components of the remaining networks (after removing nodes following different attack protocols) are computed, and also the most important robustness coefficients are computed and analyzed when metro lines disappear from the whole network.

3.5 I believe that 3.3 and 3.4 constitute a useful extension of the research activity.

Yes, we agree. This has been reflected in some way in the future work.

4. Section V:

The authors should reveal the limitations of this study, and also the plans to solve these limitations, which could be the subject of a subsequent article.

The section devoted to conclusions and future work has been modified according these suggestions.

Reviewer 2 Report

The article, in its current form, analyses the structural and robustness

characteristics of Madrid metro network. The article examines the importance

of each node in relation to the configuration (topology) of the network and

the position of the node within it and the number of lines passing through

the node.

The procedure is correct and the analysis is carried out accurately.

However, the article does not examine certain aspects, such as the number of

passengers using each node (station) as an access / egress point to the

underground network, or the number of passengers (trains) in transit on the

lines that pass through each node.

The importance of the node depends not only on the position within the

network, but also on the intensity of traffic that passes through it.

It would be interesting, for example, to know how many people would suffer

damage (delays, inability to travel) in the event of failure of individual

nodes / lines.

It would also be interesting to learn more about the possibilities of

operation of the network, in particular the number of trains or passengers

that could continue to travel, in the event of failure of a node or line;

useful indicators could be constituted, for example, by the number of

passengers directly or indirectly damaged by the failure of a single node,

or by the number of passengers not influenced by the failure.

it is necessary to give an adequate definition of the following terms, for readers who are not familiar with the subject:

L-Space topology (line 24)

Reduced L-Space (line 42)

P-Space, C-Space (line 319)

I believe that the article, very rigorous, can be published in its current

form; however, the questions raised could be the subject of future research

work."

Author Response

The article, in its current form, analyses the structural and robustness characteristics of Madrid metro network. The article examines the importance of each node in relation to the configuration (topology) of the network and the position of the node within it and the number of lines passing through the node.

The procedure is correct and the analysis is carried out accurately. However, the article does not examine certain aspects, such as the number of passengers using each node (station) as an access / egress point to the underground network, or the number of passengers (trains) in transit on the lines that pass through each node.

The importance of the node depends not only on the position within the network, but also on the intensity of traffic that passes through it.

It would be interesting, for example, to know how many people would suffer damage (delays, inability to travel) in the event of failure of individual nodes / lines.

It would also be interesting to learn more about the possibilities of operation of the network, in particular the number of trains or passengers that could continue to travel, in the event of failure of a node or line; useful indicators could be constituted, for example, by the number of passengers directly or indirectly damaged by the failure of a single node, or by the number of passengers not influenced by the failure.

It is necessary to give an adequate definition of the following terms, for readers who are not familiar with the subject:

L-Space topology (line 24)

Reduced L-Space (line 42)

P-Space, C-Space (line 319)

I believe that the article, very rigorous, can be published in its current

form; however, the questions raised could be the subject of future research

work.

RESPONSE TO REVIEWER 2:

Thank you very much for your comments!

Yes, you are right! Obviously, it is necessary to take into account additional information (such as the number of passengers, the number and frequency of trains, etc.) to obtain a more complete description and analysis of the reliability of Madrid metro network. This will be the next step in our research and this has been explicitly reflected in the “Conclusions and future work” section.

In the revised version of the manuscript (third paragraph of the Introduction) the notions of reduced L-space, P-space and C-space have been briefly explained.

Reviewer 3 Report

The paper is well-written and very interesting.  It touches important topic of metro networks as the most efficient mean of transport in sustainable smart cities. Authors provided the complex network analysis with the necessary scientific tools to perform both, quantitative and qualitative analysis of metro networks.  Authors provided a literature review focused on procedures or tools to find more efficient way for development public transport systems especially they focused on Madrid metro network which is the largest in Spain (the number of stations (243), its total length (294 km), the number of passengers - during 2017 just 626.4 million people used it).

The main structural and topological characteristics, and robustness features of Madrid metro network are studied in proper way – mathematical model, experiments with the model and the results were presented in tabular and graphical forms.

-      I suggest to define „vj” and  „vj” in the set V – in later analyses, these variables are used,

Author Response

The paper is well-written and very interesting.  It touches important topic of metro networks as the most efcient mean of transport in sustainable smart cities. Authors provided the complex network analysis with the necessary scientic tools to perform both, quantitative and qualitative analysis of metro networks.  Authors provided a literature review focused on procedures or tools to find more efcient way for development public transport systems especially they focused on Madrid metro network which is the largest in Spain (the number of stations (243), its total length (294 km), the number of passengers - during 2017 just 626.4 million people used it).

The main structural and topological characteristics, and robustness features of Madrid metro network are studied in proper way – mathematical model, experiments with the model and the results were presented in tabular and graphical forms.

-      I suggest to define „vj” and  „vj” in the set V – in later analyses, these variables are used,

RESPONSE TO REVIEWER 3

Thank you very much for your valuable comments!

In the new version of the manuscript the notation vj (which represents the j-th node/station of the metro network) has been detailed explained.

Round  2

Reviewer 1 Report

This version of the paper has a great improvement, and I am almost satisfied with the authors' reaction to my comments. However, I have two questions as follows.

1. Figure 15: When the network was applied by the (iii) betweenness-based removal strategy, which the node with the highest betweenness centrality is first deleted from the network, why the yellow line raised at the beginning.

2. Figure 17: It is quite difficult to see the result. The heat map should show the valley more clearly.

Author Response

Comments of Reviewer 1 (Round 2)

Response to Reviewer 1

First of all, we want thank you very much for your comments!

1. In the new version of the manuscript the following paragraph has been added (it appears in blue in page 21):

When one node/station is removed, the resulting network could have the same number of connected components or not (the number of connected components increases). In the first case, if the removed node has high centrality values (degree or/and betweenness) the average betweenness of the network usually increases (since the average number of the paths between any pairs of nodes decreases significantly) –this is exactly what happens in the begining-. On the other hand, if the node removed has similar centrality coefficients that the remaining nodes in the network, the average betweenness is slightly decreased. In the second case, when more connected components appear, the average betweenness decreases. This behavior is similar to the exhibited by, for example, Shaghai [Zhang et al., 2011].

Please, note that a new reference has been added.

2. Figure 17 has been modified according to your suggestion. Specifically, two lines standing for the elimination of both, line 6 and line 10, are added.
